# Rice Husk Research: From Environmental Pollutant to a Promising Source of Organo-Mineral Raw Materials

**DOI:** 10.3390/ma14154119

**Published:** 2021-07-23

**Authors:** Baimakhan Satbaev, Svetlana Yefremova, Abdurassul Zharmenov, Askhat Kablanbekov, Sergey Yermishin, Nurgali Shalabaev, Arsen Satbaev, Vitaliy Khen

**Affiliations:** 1RSE Astana Branch, National Center on Complex Processing of Mineral Raw Materials of the Republic of Kazakhstan, Nur-Sultan 010000, Kazakhstan; fnc-astana@mail.ru (B.S.); nurgali_s@bk.ru (N.S.); arsen_satbayev@mail.ru (A.S.); 2National Center on Complex Processing of Mineral Raw Materials of the Republic of Kazakhstan RSE, Almaty 050036, Kazakhstan; jarmen56@mail.ru (A.Z.); kablanbekov_as@mail.ru (A.K.); esv-ret@mail.ru (S.Y.); 3Scientific and Technical Society (STS), KAHAK, Almaty 050010, Kazakhstan; vitalyikhen@list.ru

**Keywords:** rice husk, IR, SEM, TA, TPD-MS, EPR, TEM, organo-mineral raw materials

## Abstract

Rice husk is a large-tonnage waste left from rice production. It is not subject to humification and therefore becomes a serious environmental pollutant. Due to the presence of two essential elements—carbon and silicon—in its composition, rice husk is a promising organo-mineral raw material. The known methods for processing of rice husk are associated with the formation of even more aggressive waste. The creation of a waste-free technology for processing this plant material requires a detailed study. Rice husk of Kyzylorda oblast was studied using IR, SEM, TA, TPD-MS, EPR, and TEM methods. It was determined that under a temperature up to 500 °C, the ligno-carbohydrate component of rice husk decomposes almost completely. Three main peaks are recorded during the decomposition: hemicellulose at 200 °C, cellulose at 265 °C, and lignin at 350–360 °C. This process is endothermic. However, above of 300 °C the exothermic reactions associated with the formation of new substances and condensation processes in the solid residue begin to prevail. This explains the increase in the concentration of paramagnetic centers (PMCs) in products of rice husk carbonization in the range of up to 450 °C. Further increase in temperature leads to a decrease in the number of PMCs as a result of carbon graphite-like structures formation. The silicon–carbon product of rice husk carbonization (nanocomposite) is formed by interconnected nanoscale particles of carbon and silicon dioxide, the modification of which depends on the temperature of carbonization. The obtained data allow management of the rice husk utilization process while manufacturing products in demand based on ecofriendly technologies.

## 1. Introduction

Within the last few years, works on plant waste processing have multiplied all over the world, creating a wide range of products to be used in various areas of economic activity. Kazakhstan too is focusing on environmental protection and ecological development [1,2]. The “*Birge—taza Qazaqstan*” campaign is being carried out, aiming to foster environmental values in society and cultivate a caring attitude toward nature. The country’s leadership is keen to develop agribusiness, inevitably contributing to the growth of vegetable agricultural waste. This strongly calls for the implementation of the country’s Green Economy Concept [3]. Since Kazakhstan is a rice-growing country, it is now experiencing the problem of efficient processing of rice production waste—rice husk, which is globally one of the major environmental pollutants [4]. It is known that the performance characteristics of finished materials and the scope of their application largely depend on the technological parameters of processing of raw materials. Hence, as it was revealed by an earlier review of the literature [5], the studies of rice husk are mainly devoted to the research of conditions and operating parameters of processes of its refining. There are positive results of approbation of rice husk and products of its processing in different areas of practical use [5,6]. The two most common of them are the production of activated carbon [4,7,8,9,10] and the extraction of silicon dioxide as a source of silicon for the production of pure metal and its compounds, concrete, cement, and refractory ceramics [11,12,13,14,15,16,17,18,19,20].

However, the large number of methods of processing rice husk, proposed by various researchers, does not eliminate the problem of its utilization. In Kazakhstan, industrial processing of this waste is not implemented, and rice husk continues to accumulate. Due to a number of environmental and economic reasons, other countries are holding back the organization of such industrial enterprises [5]. Firstly, the majority of the proposed methods for rice husk processing lead to the formation of new waste. These are either the acidic effluents generated during the cleaning of rice husk, toxic gases released during its incineration, or dumps of finely dispersed ash [21]. Secondly, they do not ensure the profitability of production [5]. It is obvious that a technology is needed that would ensure the maximum utilization of this agricultural waste. However, the creation of an effective technology requires a comprehensive study of not only the recycling process, but also of the raw materials themselves. As a rule, the research in the field of rice husk processing is traditionally limited to determining its composition.

Consequently, the present work is devoted to the study of Kyzylorda oblast rice husk structure by the methods of IR spectroscopy (IR) and scanning electron microscopy (SEM), determining the qualitative and quantitative composition of its organic and mineral components, the study of the decomposition of rice husk by the method of thermal analysis (TA) while determining the role of the constituent components in this process by the method of temperature-programmed desorption mass spectrometry (TPD-MS), the study of the paramagnetic centers formation in the course of rice husk carbonization and the carbonization temperature influence on it, and the study of the supramolecular structure of carbonized rice husk. The obtained data make it possible to provide insight into rice husk as an organo-mineral raw material containing two important elements—carbon and silicon. The optimum decomposition temperature for rice husk has been identified. It is shown that the destruction of rice husk occurs as a result of the disintegration of energetically weak links and removal of easily moving groups with an increase in concentration of paramagnetic centers. The temperature ranges for endo- and exothermic processes leading to the formation of graphite-like structures on the basis of carbon-containing components have been identified. The dynamics of the transformation of silicon dioxide from one form to another with an increase in the temperature of heat treatment of rice husk have been traced. Such data make it possible to effectively manage the processes of using rice husk to produce a silicon–carbon nanocomposite with a unique structure or as an independent ingredient for the production of in-demand products.

## 2. Materials and Methods

### 2.1. Materials

Rice husk from different rice-growing farms of Kyzylorda oblast was used as the object of study. The combined batch of rice husk was washed with cold distilled water at a ratio of 1:5 in order to remove the residual flour and dust. Washed rice husk was separated from the liquid by filtering through a mesh with cell size of 5 mm and left on the mesh until an air-dry condition was reached. Rice husk in an air-dry condition was put into the drying oven, where it was kept at a temperature of 105 °C until it reached a constant weight.

### 2.2. Methods of Analysis

#### 2.2.1. Infrared Spectroscopy

IR spectra were recorded on a Specord M80 spectrophotometer (Carl Zeiss, Jena, Germany) in the form of press tablets with KBr in the range of 4000–400 cm^−1^. The IR spectrum presented is the average of three measurements. Mineralogical analysis was performed by comparing the obtained IR spectra with correlation diagrams of group frequencies, as well as with reference IR spectra of monominerals [22,23].

#### 2.2.2. Scanning Electron Microscopy

Scanning electron microscopy and electron probe microanalysis were performed on a Superprobe 783 microanalyzer (JEOL, Tokyo, Japan). Analyses and photography of secondary and backscattered (composition) electrons were performed by using an INCA Energy Dispersive Spectrometer (Oxford Instruments, London, England). To avoid the formation of a charge on the analyzed materials, which are capable of deflecting the electron beam, the samples were precoated with a thin structureless gold film in a fine coat ion sputtering apparatus (JEOL, Tokyo, Japan). To clarify the distribution of individual elements, footage of characteristic X-ray radiation of corresponding elements was taken.

#### 2.2.3. Determining Rice Husk Composition

Rice husk composition determination was performed as per the methods described in [24]: the cellulose content was identified by the method of Kurschner and Hoffer, using a nitric acid alcoholic solution. For comparison, the cellulose amount was identified by determining hardly hydrolyzable polysaccharides using 80 wt.% sulfuric acid. The hemicellulose content was identified by determining easy hydrolyzable polysaccharides using 2 wt.% hydrochloric acid. The quantitative identification of lignin was performed using 72 wt.% sulfuric acid as per Komarov’s modification. The total quantity of extractive substances was identified by treatment with an alcohol–benzol mixture in a Soxhlet apparatus, as well as with hot water. The content of mineral components in rice husk was identified based on silicate chemical analysis.

#### 2.2.4. Thermal Analysis

Thermal analysis was performed on a Hungarian Paulik F.–Paulik J.–Erdey L. system derivatograph Q-1500D (MOM, Budapest, Hungary). The smooth heating of the sample was performed to a temperature of 1000 °C with a temperature increase rate of 12 °C min^−1^ in the atmosphere of the exhaust gas.

#### 2.2.5. Temperature-Programmed Desorption Mass Spectrometry 

TPD-MS was performed on a MX-7304A monopole mass spectrometer (Electron, Sumy, Ukraine) with electron ionization, which was re-equipped for thermal desorption measurements. A sample weighing 1–20 mg was placed on the bottom of a quartz-molybdenum ampoule and before the start of the experiment was pumped out at room temperature to a pressure of ~5 × 10^−5^ Pa. The programmed linear heating of the sample was performed at a speed rate of 0.15 °C s^−1^ to a temperature of ~750 °C. The volatile products of thermolysis directly entered the ionization chamber of the mass spectrometer through a high-vacuum valve 5.4 mm in diameter, ionized and fragmented under the action of electrons. After the separation by masses in a mass analyzer, the intensity of the ion current of the products of desorption and thermolysis was recorded by a VEU-6 secondary electron multiplier (“Gran” Federal State Unitary Enterprise, Vladikavkaz, Russia). The registration and analysis of mass spectra and curves of dependence of the pressure of volatile destruction products on the temperature of the sample (P-T) were performed by the automated computer-based data recording and processing system. The registration of mass spectra was performed in the range of 1–210 amu. During the TPD-MS experiment, about 240 mass spectra were recorded. During the thermal desorption experiment, the samples were heated rather slowly and the rate of evacuation of volatile thermolysis products was high, allowing us to neglect the diffusion effects; therefore, the intensity of the ion current was proportional to the rate of desorption.

#### 2.2.6. EPR Spectroscopy

EPR spectroscopy of the initial and carbonized rice husk in the range starting from 200 °C to 800 °C with a step of 50 °C was performed on an upgraded EPR IRES-1001-2M homodyne spectrometer (JEOL, Tokyo, Japan), which operates in the 3 cm wavelength range, at room temperature and optimal conditions for registration of spectra: a magnetic field of 120 Oe with a microwave power of 16 mW and modulation amplitude of 1 Oe. These conditions of spectra recording were chosen as optimal because the microwave power value eliminated the effects of EPR saturation and the magnetic modulation amplitude eliminated broadening of the EPR line. EPR spectra were recorded between 3 and 4 components of the reference sample, which were ions Mn^2+^ in MgO. G-factor and an EPR line width of tested samples were determined using known reference sample parameters. The concentration of free radical states of tested samples was calculated by comparison of areas of their spectra and the calibrated reference sample (third line of EPR spectrum ions Mn^2+^ in MgO).

#### 2.2.7. Transmission Electron Microscopy (TEM)

Transmission electron microscopy studies were performed on different instrumental equipment:-On the EM-125K device (Sumy electronic devices plant, Sumy, Ukraine) by the method of direct observation of translucency by using the microdiffraction. The samples were prepared by the method of dry preparation, i.e., by the method of dry application of the agent to a collodion backing film and by the method of one-stage carbon replicas with extraction. During the microdiffraction studies, the photographing of diffraction patterns was performed.-On the Transmission Electron Microscope Philips EM 301 (Philips, Amsterdam, Netherlands) at an accelerating voltage of 80 kV in the range of electron microscopic magnifications of 13–80 thousand times. The images were recorded with an Olympus C-3040 digital camera, which was operated via computer using the Image Scope M software (Systems for microscopy and analysis (SMA), Moscow, Russia). The objects were prepared as follows. A small quantity of the sample was ground in an agate mortar. The resulting powder was applied to an object copper grid previously coated with an amorphous carbon backing film. The object grid with the applied sample was fixed into the microscope object holder and inserted into the microscope column [25].

#### 2.2.8. Rice Husk Carbonization and Extraction of Silicon Dioxide

Rice husk carbonization for further research was performed in a shaft furnace SSHOL-8/11 (Tula-Term, Tula, Russia) in an atmosphere of exhaust steam gases. For this purpose, the reactor was filled with 200 g of rice husk, hermetically sealed with a cap that has a tube for exhaust gas removal and placed into the working area of the furnace. Heating to a specified temperature (in the range of 200 to 1000 °C with a step of 50 °C) was performed at a speed rate of 15 °C min^−1^, keeping the sample at this temperature for 30 min. 

In order to extract silicon dioxide, rice husk carbonized at 650 °C was heat-treated at 800 °C in the open air in order to burn off the carbon. The process was performed in a rice husk carbonization unit. During the process, the reactor was closed with cap, which, in addition to the gas outlet tube, had the air inlet tube that supplied the air inside of the reactor through the whole layer of carbonized rice husk. The process was conducted for 1 h. The yield of the resulting product of yellow-white color in terms of rice husk was equal to 14.8 wt.%. 

The separation of carbon and silicon dioxide in carbonized rice husk was performed with the help of a sodium hydroxide solution with a concentration of 70 g dm^−3^ at a solid:liquid ratio (S/L) of 1:15. The content of silicon dioxide in the carbon residue was equal to 2–3 wt.%.

## 3. Results

### 3.1. Infrared Spectroscopy Study of Rice Husk

As can be seen from the IR spectrum (Figure 1), rice husk of Kyzylorda region has a complex functional composition. In general, the spectrum is characterized by band widening. This may be due to various reasons: irregular intra- and intermolecular interactions, overlapping absorptions of different types of vibrations, and absorption at frequencies slightly different from each other. As a result, the bands merge, and many of them do not have independent peaks but are recorded as a shoulder on more intense lines. All of this creates certain difficulties in the spectrum interpretation.

In the IR spectrum of rice husk, a series of bands of different intensity are observed with maxima at 3410, 2975, 1640, 1460, 1425, 1375, 1320, 1160, 1065, 1035, and 895 cm^−1^. They can be caused by vibrations of functional groups of cellulose, which is the main component of plant tissues [26]. Cellulose is a polysaccharide whose molecules (C_6_H_10_O_5_)_n_ are long chains with a spatially regular structure. These chains consist of β-D-glucose (β-D-glucopyranose) links connected by glucoside bonds 1–4 [24].

Bands at certain wavenumbers can be caused by vibrations of the functional groups of different compounds that make up rice husk. For example, the band at 1160 cm^−1^ can also belong to the vibrations of the C–O bonds of oxygen-containing groups of hemicelluloses. Hemicelluloses are found in the cell walls of plants and are composed of polysaccharides containing elementary units of five to six carbon atoms. Most of the hemicelluloses are not homogeneous polysaccharides, but mixed. Mixed polysaccharides are composed of various monosaccharide residues linked by glucoside bonds at various positions. Therefore, a broad band with a maximum at 3410 cm^−1^ can correspond to stretching vibrations of hydroxyl groups included in hydrogen bonds of hemicelluloses as well [27].

The bands with maxima at wavenumbers 1595 and 1512 cm^−1^ are typical for skeletal vibrations of aromatic rings. Their presence indicates the presence of lignin [28]. Lignin is a natural polymer built from the structural elements of C_6_C_3_ oxygen derivatives of phenylpropane produced from carbohydrates [29].

According to paper [28], the band at 1640 cm^−1^ can also be caused by vibrations of carbonyl groups conjugated with condensed nuclei. The peak with a maximum at 1725 cm^−1^ is due to vibrations of conjugated aldehyde and ketone groups and non-conjugated carbonyl and carboxyl groups [30]. The opinions of researchers differ on the interpretation of the absorption band at 1275 cm^−1^. Some believe [31] that this band is due to asymmetric stretching vibrations of C-O-Si bonds in the methoxyl groups of lignin. Others attribute it mainly to bending vibrations of methylene CH_2_ groups [32].

The bands at 1085, 795, and 465 cm^−1^ are characteristic bands of the silica component [22]. No clear absorption bands corresponding to the presence of C-O-Si or Si-CH_3_ bonds in rice husk were recorded. However, the intense band with a maximum at 1085 cm^−1^ has shoulders at 1070 cm^−1^ and 1040 cm^−1^, as well as diffuse absorption bands at 1275 cm^−1^ and 1225 cm^−1^, typical for the valence vibrations of C-O-Si or Si-CH_3_, respectively [33].

### 3.2. SEM Study of Rice Husk

Figure 2a shows a longitudinal section of rice husk, characterizing its shape. Rice husk has a base formed by heterogeneous fibers. A wave-like “shell”, the inner filling of which is represented by a loose mass of plant tissue, is held at the base. Numerous cracks give visible friability to the specimen. Dispersed particles, possibly introduced as a result of mechanical destruction of rice husk, are observed in the deep cracks. Figure 2 shows the distribution of carbon (Figure 2b), silicon (Figure 2c), and oxygen (Figure 2d) in a longitudinal section (Figure 2a) of rice husk. The image was obtained in backscattered electrons (in the characteristic emission of C, Si, and O, respectively). The distribution of elements in the figure is characterized by the cluster of light dots. As shown in Figure 2c, silicon is predominantly located in the outer surface layer and is also localized in some places of the plant tissue. Local accumulations on the inner layer are also explained by the presence of destroyed surface layer particles. The distribution of oxygen (Figure 2d) follows the shape of silicon and carbon. However, the image contrast in the case of carbon and oxygen is lower in comparison to silicon. This may be explained as follows. It is known that the higher the atomic number of the element and the greater the difference between the atomic numbers of the elements being compared, the higher the probability of backscattering and the higher the image contrast. The presence of silicon dioxide in rice husk composition distinguishes it from other plant materials.

After rice husk thermal treatment, a redistribution of chemical elements was registered by SEM (Figure 3). As a result of the destruction of the rice husk organic component, fusion of the material occurs (Figure 3a). Deep channels are formed in the places of burnout of longitudinal fibers and along the contour of the “shell”. A peculiar carbon matrix is formed (Figure 3b). It is evenly filled with a silicon-containing phase (Figure 3c). As can be seen from the accumulation of white dots in Figure 3d, the oxygen distribution corresponds to the silicon distribution form. Thus, the thermal destruction of rice husk leads to the formation of a silicon-carbon composite. 

### 3.3. Rice Husk Composition

Due to the intricate structure of plant tissue, all of its constituent components are very closely related to each other. This makes it difficult to separate them. Thus, cellulose in plant cell walls is closely related to hemicelluloses, lignin, and extractive substances, while lignin is partially penetrating the cellulose microfibrils. Some hemicelluloses (cellulosans) form associates with cellulose and cannot be removed from plant tissue without noticeable damage to the cellulose itself. Therefore, the techniques described in Section 2.2.3 were used to perform the most accurate quantification of rice husk composition.

When treating rice waste with a mixture of concentrated nitric acid and ethyl alcohol, the lignin is nitrated and partially oxidized, and transferred to the alcohol solution. Hemicelluloses are hydrolyzed for the most part. Alcoholic medium moderates the oxidizing and hydrolyzing effect of nitric acid on cellulose. The cellulose content in the studied sample of rice husk, determined by this method, was 33 wt.%. This figure agrees well enough with the amount of cellulose (30 wt.%) established by the method of determination of hard-to-hydrolyze polysaccharides using 80 wt.% sulfuric acid. Hemicellulose content was determined according to the method of determination of easily hydrolyzable polysaccharides using 2 wt.% hydrochloric acid. Their quantity in the composition of rice husk was 18 wt.%. Since the composition of extractive substances is extremely diverse and quantitative isolation of individual components is rather complicated, the total amount of extractive substances soluble in the alcohol–benzene mixture (i.e., resins) and hot water was determined in the composition of rice husk, which was 2.0 wt.% and 6.1 wt.%, respectively. Unlike polysaccharides (cellulose and hemicelluloses), which are hydrolyzed to simple sugars, lignin is resistant to the action of mineral acids. Therefore, its content was determined using 72 wt.% sulfuric acid at room temperature on rice husk previously deresinated with an alcohol and benzene mixture. The reaction mixture was then diluted with water and boiled. The amount of lignin determined by this way was 26 wt.%.

By the silicate chemical analysis scheme, it was found that rice husk contained mineral components such as silicon dioxide (~14.0 wt.%), calcium oxides (~0.2 wt.%), magnesium (~0.1 wt.%), iron (~0.02 wt.%), aluminum (<0.1 wt.%), potassium (~0.4 wt.%), sodium (~0.03 wt.%), and other elements at impurity levels.

### 3.4. Investigation of Rice Husk Thermal Degradation Process by Thermal Analysis

Thermal analysis of rice husk in the region of 50–150 °C captures on the DTA curve an endo effect related to the loss of free water (Figure 4). The decomposition of rice husk in the exhaust gas atmosphere ends at 770 °C, as can be seen on the TG curve. The DTG curve shows that the bulk of rice husk decomposed with the highest rate at 238 and 265 °C. This process runs with heat absorption, since two blurred endo effects are fixed on the DTA curve at these temperatures. The exothermic effect recorded on the DTA curve at 300 °C is caused by the formation of new substances and condensation processes in the solid residue. It is alternately replaced by endo (450, 550 °C) and exo (500, 600 °C) effects caused by structuring and burnout of carbonaceous residue. Rice husk mass loss reached 79.5%. The excess mass of the residue (20.5wt.%) over the mineral component in the composition of rice husk (~15wt.%) is explained by the presence of carbon formed during condensation–destruction processes and subsequent graphitization (since TA was performed in the atmosphere of the exhaust gas) and is probably firmly bound to silicon dioxide. 

### 3.5. Study of Rice Husk Pyrolysis by TPD-MS

Analysis of correlation curves between the pressure of volatile degradation products of plant materials and heating temperature (P/T) showed (Figure 5) that rice husk decomposes in the temperature range of 150–500 °C, with the maximum decomposition being fixed at T_max_ ~265 °C. The data obtained are in good agreement with the TA results. Comparison of Figure 5 and Figure 6 indicates that the release of the bulk of rice husk pyrolysis products at T_max_ ~265 °C is caused by cellulose degradation due to the desorption of pyran and furan derivatives (*m*/*z* 98). The formation of pyran derivatives is caused by the dehydration of elementary links in the pyranose form; the formation of carboxylic acids followed by decarboxylation promotes the formation of furan derivatives; at the same time, the appearance of aromatic structures at this temperature indicates the sequential course of intermolecular and intramolecular aldol condensation reactions [34]. The pyrolysis stage with T_max_ ~350 °C occurs due to the degradation of aromatic compounds of lignin, which proceeds in a wider temperature range with the formation of phenol (*m*/*z* = 94, T_max_ = 350 °C), pyrocatechol (*m*/*z* = 110, T_max_ = 290–330 °C), cresols (*m*/*z* = 107, T_max_ = 365 °C), tropylium ion, C_7_H_7_^+^ (*m*/*z* = 91, T_max_ = 340 °C), 4-vinyl-methylguaiacol (*m*/*z* = 164, T_max_ = 290 °C), benzene (*m*/*z* = 78, T_max_ = 360, 545 °C), naphthalene (*m*/*z* = 128, T_max_ = 520 °C), and 4-vinylphenol (*m*/*z* = 120, T_max_ = 220, 290 °C). The formation of these compounds is caused by thermal transformations of corresponding aromatic links and functional groups of lignin [35]. The stage with T_max_ ~200 °C corresponds to the destruction of hemicelluloses [36]. However, due to their close relationship with other plant matter constituents, the above compounds predominate in the pyrolysis products at 150–250 °C (Figure 6b). At the same time, the ion with *m*/*z* = 60 (HOCHCHOH^+^) observed in Figure 6a is known to be the most intense marker ion in the mass spectra of carbohydrate pyrolysis products and is usually detected already at 150 °C [37,38].

### 3.6. Investigation of Rice Husk Structural Changes during Carbonization by EPR Spectroscopy

When studying the structural changes of rice husk during heat treatment up to 800 °C using EPR spectroscopy, it was found that the material already exhibits an EPR signal at room temperature. This signal can be caused by the formation of free radicals under mechanical impact during rice husk grinding. The parameters of the EPR spectra are presented in Table 1. For illustrative purposes, the dependence of paramagnetic centers (PMCs) content on the processing temperature of rice husk is shown in Figure 7. The concentration of paramagnetic centers reaches a maximum value (1000 × 10^16^ spin g^−1^) at 450 °C. There is a general tendency of reducing the width of the EPR line while increasing temperature of heat treatment, as it follows from Table 1. However, despite general narrowing of the line, it broadens at certain temperatures (400 and 450 °C). Obviously, at these temperatures two kinds of paramagnetic centers are formed (free radicals and clusters as a result of closure of the former) with similar values of g-factor, the superposition of their signals leads to the broadening. 

Comparing the pattern of PMCs concentration changes as the plant raw material processing temperature increases with the results of TA, IR spectroscopy, and TDS-MS, it can be concluded that the growth of PMCs amount when heating rice husk to 450 °C is caused by an increase in the concentration of free-radical states (FRS) as a result of splitting of energy-weak bonds and removal of easily mobile groups. Carbonization of plant material is inevitably accompanied by the formation of condensed carbon rings that form a graphite-like structure. This leads to a decrease in the value of the PMCs index. Consequently, on the ascending part of the paramagnetic centers concentration dependence on the treatment temperature (Figure 7), the ΔH decrease (Table 1) is explained by intensification of exchange interactions in the FRS spin system as their concentration increases, while on the descending part—by a decrease in the dipole–dipole interaction and appearance of delocalized π electrons in graphite-like structures. However, treatment of rice husk carbonized at 650 °C with a sodium hydroxide solution causes PMCs concentration to increase to 1.1 × 10^19^ spin g^−1^. This is obviously associated with the removal of silicon dioxide from carbonizate due to the breaking of C-SiO_2_ bonds and formation of a large amount of FRS. In the studied temperature range of rice husk heat treatment, gradual decrease of g-factor values also occurs, which approach the free electron g-factor value (g = 2.0023) in graphite structures. Thus, we can conclude that rice husk structural transformations during heat treatment undergo a free radical formation stage followed by the formation of hexagonal meshes of cyclically polymerized carbon. 

### 3.7. Characteristics of Carbonized Rice Husk Supramolecular Structure 

The presence of particles of different morphology and sizes was recorded in rice husk carbonized at different temperatures (600, 650, and 1000 °C) using transmission electron microscopy (Figure 8). Mainly, there are lamellar translucent and dense particles (Figure 8a); aggregates formed by translucent particles of round or oval shape of 15–20 nm and larger (Figure 8b). However, more attention should be given to hybrid structures, which are a combination of two phases (Figure 8c,d): a lamellar formation of one phase permeated by another denser dispersed (10–20 nm) phase. Moreover, the hybrid particles can be porous (pore size 15–20 nm, Figure 8c). Particles whose microdiffraction pattern (Figure 8e,f) is represented by rings (0.337; 0.210; 0.122 nm), symmetric reflexes (0.46; 0.406; 0.337; 0.251; 0.245; 0.236 nm) and broad rings with a set of interplanar distances of 0.253; 0.212; 0.152; 0.121 nm were determined in the rice husk carbonizate by microdiffraction studies along with amorphous particles. According to paper [39], this indicates the presence of the following carbon and silica-containing phases: C (26–1080), H_2_Si_14_O_29_ ⋅5.4H_2_O (31–584), SiC/Unnamed mineral, syn. (29–1129).

To study the structure of silica, carbonized rice husk was heat-treated at 800 °C in open air to burn out the carbon component. Figure 9 shows clusters of dense particles of 20 nm and larger (Figure 9a) and translucent rounded particles of 20 to 100 nm in diameter (Figure 9c). The microdiffraction pattern from dense particles is represented by reflexes with a set of interplanar distances of 0.432; 0.375; 0.286; 0.261; 0.231; 0.207; 0.154; 0.15 nm (Figure 9b), corresponding to a mixture of SiO_2_ (12–708), SiO_2_ (18–1169), and H_2_Si_2_O_5_ (27–606) [39]. The appearance of a 0.428 nm diffuse ring in the microdiffraction pattern from the translucent particles (Figure 9d) indicates the nucleation of the SiO_2_/Tridymite-20H, syn. (14–260).

## 4. Discussion

Analysis of rice husk and products of its heat treatment showed the following. Rice husk is an organo-mineral material. As a large-tonnage waste of rice production, it is a large-scale environmental pollutant, since due to the presence of silicon dioxide it is not subject to the humification process. At the same time, the combined presence of carbon and silicon dioxide in its composition opens up broad prospects for its use as a valuable raw material. The value of rice husk lies in ensuring the formation of a unique silica–carbon structure of the materials produced from it.

The state of carbon and silicon dioxide in the resulting products depends on the processing conditions of plant raw materials. It is possible to get carbon with an amorphous or graphite-like structure. The situation is similar to silicon dioxide. Starting as amorphous, various crystalline forms of silicon-containing products can be obtained: cristobalite, tridymite, and even silicon carbide as a result of the interaction of silicon dioxide with the resulting carbon. Carbonized rice husk is made up of different-shaped particles. The hybrid structures formed by particles of carbon and silicon dioxide have the greatest interest among them.

Due to the fact that silicon dioxide is connected strongly with organic components in rice husk after the thermal destruction of plant tissue, it does not form an isolated structure but rather has a strong interaction with carbon (up to chemical bond), and as a result it builds a nanocomposite ensemble. Obviously, being formed inside the plant cell in the form of soluble silicic acid (which the presence of was registered by a TEM microdiffraction technique even in carbonized rice husk), silica diffuses through the membranes of plant tissue to its outer surface, causing silicification of the cellulose scaffold, which was confirmed by the results of SEM study of the elements distribution in the rice husk structure. Some of it remains in the inner layer. In both cases, the formation of bonds between silica tetrahedra, carbohydrates, and lignin is possible. According to TA and EPR spectroscopy, the temperature of 650 °C appears to be the necessary and sufficient temperature for the carbonization process of rice husk, although according to TPD-MS data, rice husk decomposes in the range up to 500 °C in three stages corresponding to the decomposition of the main components (hemicellulose, cellulose, lignin). At 650 °C, the destruction of the initial raw materials is completed, as shown by TEM study of the supramolecular structure with the formation of a silicon–carbon nanocomposite in which some of the C-SiO_2_ bonds remain intact. This was confirmed by the results of different techniques used. Firstly, as it was found during TA, the residue mass after rice husk decomposition in the exhaust steam gas atmosphere (under analogous conditions carbonization of the feedstock was also performed for the study by other methods) exceeded the mass of the mineral component in rice husk due to the formation of cyclically polymerized carbon and probably SiC, as evidenced by the EPR and TEM results. A subsequent attempt to obtain carbon freed from silicon dioxide by the chemical way was unsuccessful. The residual content of SiO_2_ in carbonized rice husk treated with an alkaline solution was 2–3 wt.%. Moreover, after this treatment of the obtained product, according to EPR spectroscopy data the number of free radicals increased significantly as a result of the destruction of the C-SiO_2_ bonds present in the silicon–carbon composite. Thus, the results of studies carried out by different methods within the framework of the present work are in good agreement and complement each other.

Carbonized rice husk with a unique structure has a wide range of applications: as an active filler for elastomers [40], a reducing agent in electrothermal metallurgical processes [6], a sorbent [41], and a feed additive [6]. However, in a number of processes, it seems economically more profitable to use rice husk as raw material with the formation of the necessary compounds directly during the process of obtaining the final product. Examples of such processes are the production of new generation refractory materials by the method of self-propagating high-temperature synthesis (SHS) [42,43,44,45] and the production of plate material in the process of vapor-explosive hydrolysis without the use of any types of synthetic plastics [46].

The modern concept for the development of the refractory industry consists of the transition to the production of resource-saving refractories of a new generation, distinguished by increased environmental safety and wear resistance, as well as ensuring an increase in the quality of the final product. The feasibility of creating a new generation of refractories is due to the increasing requirements of consumers, as well as the need to improve the operating conditions of refractories and reduce energy costs in their manufacturing. For the development of refractory production, the Republic of Kazakhstan has sufficient raw materials. There are significant reserves of refractory clays, quartzite, chrome ores, small deposits of magnesite, zircon, talc-magnesite, bauxite, man-made raw materials represented by waste from the mining and metallurgical industry [47,48], as well as mineral raw materials for the production of composite materials and ceramics based on corundum, zircon, andalusite, and barite.

Carborundum (silicon carbide) is often used to increase the strength of refractory materials. As part of the present work, we recorded silicon carbide formation during the carbonization process of rice husk. This process was studied and described in detail in papers [21,49]. Therefore, it seems reasonable to introduce rice husk into the charge to produce refractories. When a specially prepared charge is heated to 950 °C, the process of self-propagating high-temperature synthesis will take place. Under the conditions of this SHS process, the organic component of rice husk will be carbonized to form nanoscale carbon. As a result of carbon interaction with silicon dioxide present in rice husk in its active form, silicon carbide will be formed. On the one hand, this will reduce or completely eliminate the consumption of silicon-containing ingredients to be introduced into the charge. On the other hand, enrichment of reaction products with highly fire-resistant (low porosity and high density) compounds will contribute to the formation of durable and dense refractory, increasing thermal durability of refractory lining to be used in chemically aggressive environments of ferrous and nonferrous metallurgy, energy, chemical industry, and construction materials production. 

At present, the issues of using plant fibers from agricultural waste in the manufacturing of building boards are quite pressing. There are a number of problems that need to be solved at the same time [50,51,52,53,54,55,56,57,58]:-Release of volatile organic substances, aldehydes, and terpenes, a small amount of which causes adverse effects on health.-Unsatisfactory physical and mechanical properties of the resulting materials, providing mainly as use for decoration and furniture production, but limiting their use in the construction industry.-Use of a large number of highly toxic and inflammable artificial organic polymers (formaldehyde, epoxy, and other resins such as binders, various types of hardeners, plasticizers, and adhesives) since almost all technologies are based on pressing plant biomass. Strengthening the mechanical properties usually requires increasing the amount of resin binder. Increasing fire resistance and improving other properties such as sound absorption, impact resistance, and thermal conductivity is associated with a more complex composition of the blend, i.e., increasing the number of ingredients and different methods of processing can increase in the cost of the finished material.

In this regard, there is independent interest to try and use rice husk for the production of building boards by the method of steam explosive hydrolysis, which determines the effective decomposition of organic raw materials [46]. The method is based on implementing hydrothermal degradation of polysaccharides by alternating the stages of pressing with charge heating and decompression to form a silicon–lignin solid residue. Plasticizing properties of lignin will allow eliminating the use of synthetic binders. The presence of silicon dioxide in the board composition will contribute to increasing its strength. This direction of research is attractive because it opens the prospect of developing the production of efficient construction boards on the basis of ecofriendly technology using renewable raw materials.

## 5. Conclusions

Rice husk of Kyzylorda region has a complex functional composition. The main components are polysaccharides (48–52 wt.%) and lignin (26 wt.%). A distinctive feature of rice husk is its high ash content due to the presence of silicon dioxide (14 wt.%). Silicon dioxide is predominantly evenly located in the outer surface layer of rice husk and in the form of local accumulations in its internal surface layer.

Thermal destruction of rice husk occurs at up to 500 °C in three stages. Hemicelluloses decompose at 200 °C. The maximum decomposition at 265 °C is caused by the destruction of cellulose. In the range of 350–360 °C, the destruction of lignin takes place. The decomposition process of rice husk is endothermic. Above 300 °C, exothermic reactions predominate due to the formation of new substances and condensation processes in the solid residue.

In the process of rice husk carbonization at 450 °C, the concentration of paramagnetic centers increases due to the splitting of energetically weak bonds and the removal of easily mobile groups. A further increase in temperature to 800 °C is accompanied by a decrease in the number of PMCs as a result of the formation of graphite-like structures since the g-factor value approaches free electron g-factor value (g = 2.0023) in graphite structures. The preferred carbonization temperature is 650 °C.

Carbonized rice husk is a silicon–carbon nanocomposite formed by nanosized particles of carbon and silicon dioxide with the presence of silicon carbide. Thanks to its unique structure, the silicon–carbon nanocomposite has a wide range of applications. The studies performed and the results obtained make it possible in the future to test rice husk as an independent charge ingredient in the preparation of refractories by the SHS method and building plates by the vapor-explosive hydrolysis method.

## Figures and Tables

**Figure 1 materials-14-04119-f001:**
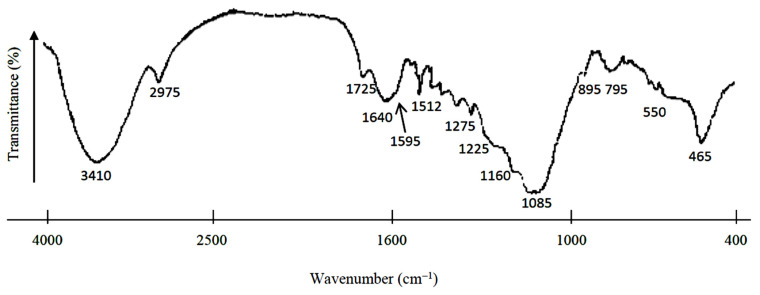
IR spectrum of rice husk.

**Figure 2 materials-14-04119-f002:**
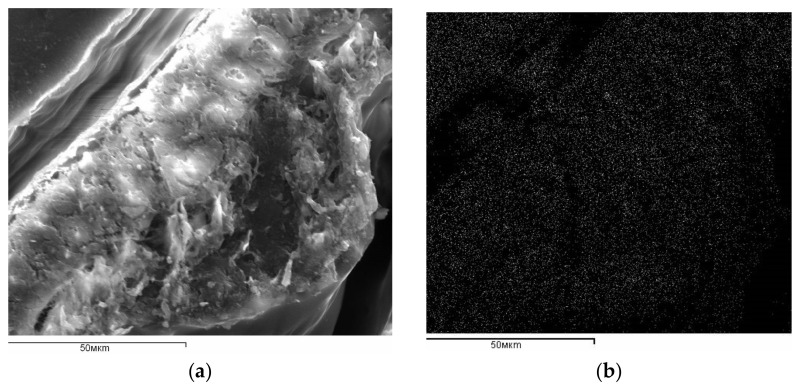
SEM micrographs of the rice husk sample: (**a**) longitudinal section of rice husk; (**b**) distribution of carbon; (**c**) distribution of silicon; (**d**) distribution of oxygen.

**Figure 3 materials-14-04119-f003:**
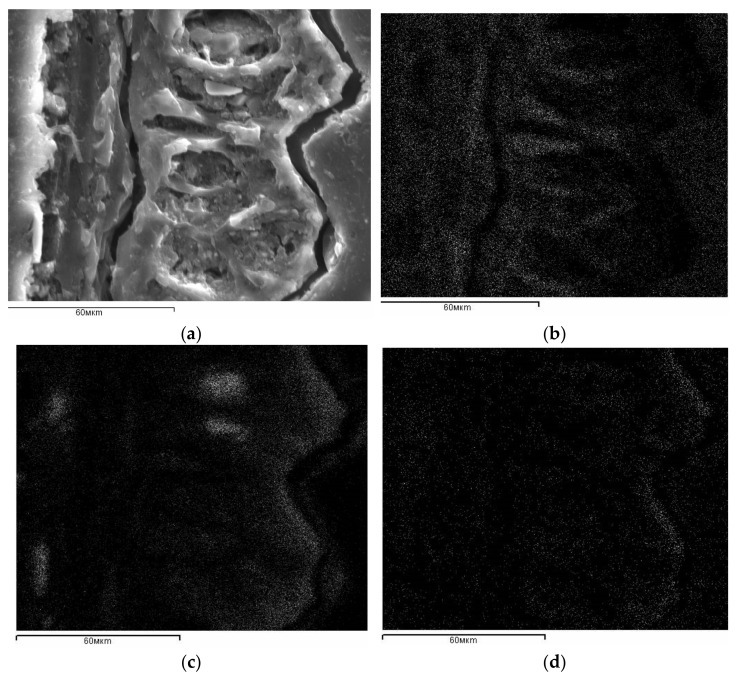
SEM micrographs of the rice husk carbonized at 650 °C sample: (**a**) longitudinal section of rice husk carbonized at 650 °C; (**b**) distribution of carbon; (**c**) distribution of silicon; (**d**) distribution of oxygen.

**Figure 4 materials-14-04119-f004:**
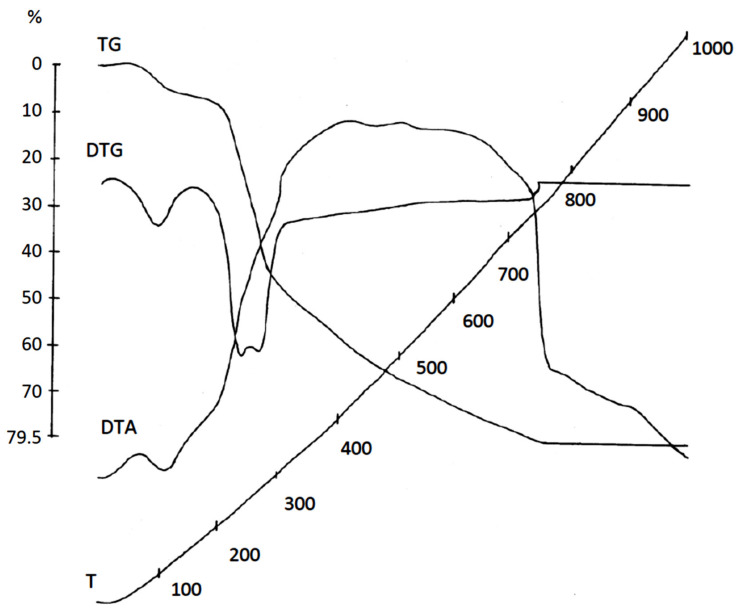
Rice husk derivatogram.

**Figure 5 materials-14-04119-f005:**
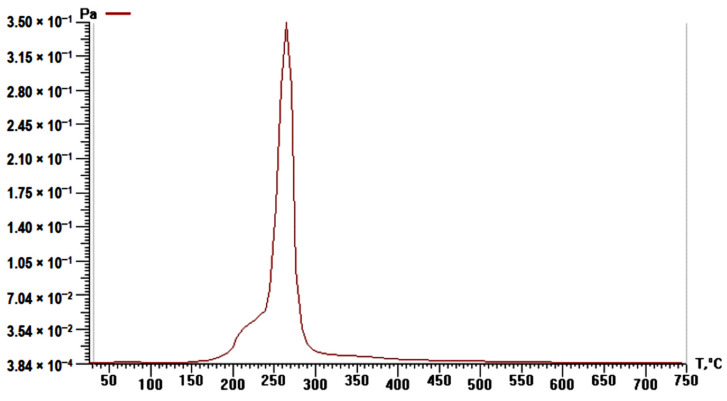
Temperature–pressure (P-T) curves of rice husk.

**Figure 6 materials-14-04119-f006:**
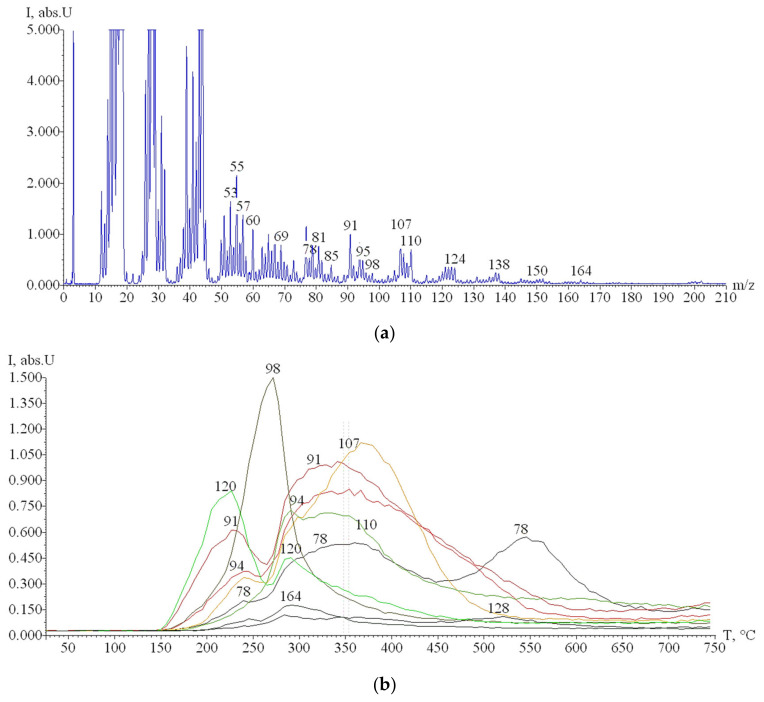
TPD-MS results: (**a**) mass spectrum of pyrolysis products of rice husk at 347 °C obtained after electron impact ionization; (**b**) TPD-curves of the ions with *m*/*z* 164, 128, 120, 110, 107, 98, 94, 91, and 78 under pyrolysis of rice husk.

**Figure 7 materials-14-04119-f007:**
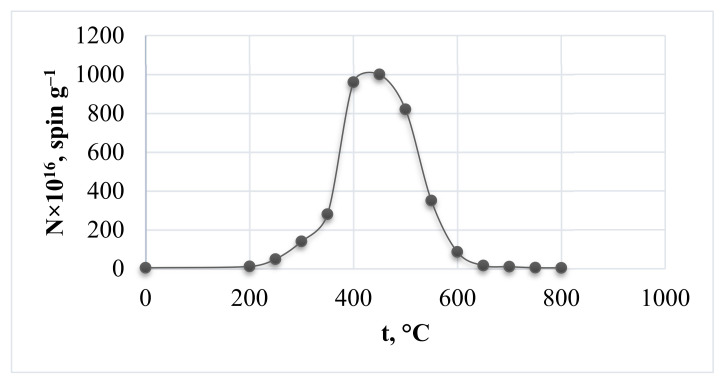
Dependence of the concentration of paramagnetic centers on the carbonization temperature of rice husk.

**Figure 8 materials-14-04119-f008:**
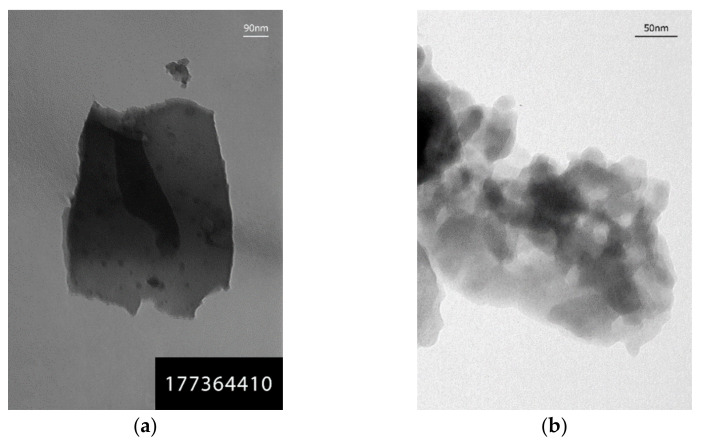
TEM micrographs and microdiffraction patterns of carbonized rise husk: (**a**) rice husk particles carbonized at 650 °C; (**b**) rice husk particles aggregates carbonized at 1000 °C; (**c**) porous hybrid formation obtained by rice husk carbonization at 600 °C; (**d**) cluster of differently shaped particles of rice husk carbonized at 650 °C; (**e**) microdiffraction patterns from the particles shown in Figure 8a; (**f**) microdiffraction patterns from the particles shown in Figure 8d.

**Figure 9 materials-14-04119-f009:**
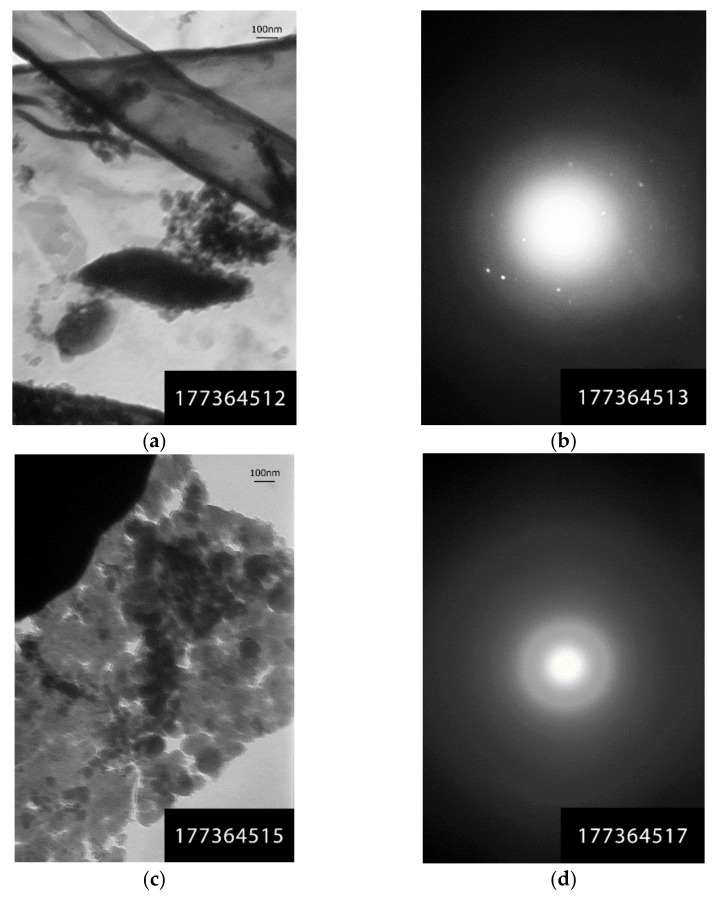
TEM micrographs and microdiffraction patterns of rice husk silica: (**a**) cluster of dense particles; (**b**) microdiffraction pattern from dense particles; (**c**) cluster of translucent particles; (**d**) microdiffraction pattern from translucent particles.

**Table 1 materials-14-04119-t001:** Parameters of EPR spectra of original and carbonized rice husk.

Sample	PMCs Concentration, spin g^−1^	EPR Line Width	g-Factor
RH	4.4 × 10^16^	8.4	2.0041
RH, 200 °C	1.1 × 10^17^	7.4	2.0039
RH, 250 °C	4.9 × 10^17^	4.9	2.0030
RH, 300 °C	1.4 × 10^18^	4.8	2.0030
RH, 350 °C	2.8 × 10^18^	4.8	2.0029
RH, 400 °C	9.6 × 10^18^	5.7	2.0027
RH, 450 °C	1.0 × 10^19^	5.6	2.0027
RH, 500 °C	8.2 × 10^18^	4.6	2.0027
RH, 550 °C	3.5 × 10^18^	4.4	2.0027
RH, 600 °C	8.6 × 10^17^	3.8	2.0027
RH, 650 °C	1.6 × 10^17^	3.3	2.0027
RH, 700 °C	9.8 × 10^16^	2.9	2.0025
RH, 750 °C	4.6 × 10^16^	2.4	2.0025
RH, 800 °C	3.9 × 10^16^	2.5	2.0023
RH, 650 °C, NaOH	1.1 × 10^19^	4.8	2.0026

## Data Availability

Data sharing is not applicable to this article.

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
