# Peer review of "Rice Husk Research: From Environmental Pollutant to a Promising Source of Organo-Mineral Raw Materials"

_materials, 2021, doi:10.3390/ma14154119_

Round 1

Reviewer 1 Report

See attachment

Author Response

Dear Reviewer,

Authors would like to thank you for useful recommendations and your time spent reviewing the manuscript. Your comments are very valuable for us. In general, we agree with your point of view. As a result, we paid attention to each of your suggestions. In our opinion, revision undertaken according to your comments really helped to improve the manuscript. Please see our response attached.

Thank you again.

Best regards,

Prof. Svetlana Yefremova

Reviewer 2 Report

In this paper, the authors investigate the processing method of rice husk. They find that thermal destruction of the bulk of the rice husk occurs up to 500°C, and it happens in three stages: at 200°C, Hemicelluloses will decompose; at 265 °C, the cellulose will be destructed, which counts the maximum decomposition; at 350-360 °C, the lignin will be destructed.  In general, the structure of the paper is clear and the results are well delivered. However, there are a few concerns that need to be addressed:

  1. The scale bar in the SEM images in figure 2 is merely recognizable. Can the authors fix this?
  2. I am not sure anything can be observed from figures 2b and 2d. I would suggest the authors repeat the imaging and take a better image.

Author Response

Dear Reviewer,

Authors would like to thank you for your reviewing and comments. We tried to clarify the issues. Figure 2 was improved. Explanation on the second question is provided in the attached response.

As for English language, spelling was checked and corrected, where we thought it was required.

Thank you again.

Best regards,

Prof. Svetlana Yefremova
